# Correlations between Achilles tendon moment arm and plantarflexor muscle architecture variables

Logan Faux-Dugan👤[ID]◉, Stephen J. Piazza*◉

Department of Kinesiology, Biomechanics Laboratory, The Pennsylvania State University, University Park, PA, United States of America

◉ These authors contributed equally to this work.
* piazza@psu.edu

**Data Availability Statement:** Data files, including folders with summaries of the results, accuracy and reliability assessments, and subject-specific data have been made publicly available. For each subject, there are stradwin files displaying muscle

## Abstract

The production of triceps surae plantarflexion moment is complex in that the Achilles tendon moment arm affects the Achilles tendon force by determining the muscle length change and shortening velocity during ankle rotation. In addition, there is evidence for associations between the sizes of muscles and their moment arm at the joints they span. These relationships between muscle architecture and tendon moment arm ultimately affect the muscle's force generating capacity and are thus important for understanding the roles played by muscles in producing locomotion. The purpose of this study was to investigate *in vivo* the relationship between architecture of two plantarflexors and the Achilles tendon moment arm in a healthy adult population. Ultrasound-based measurements were made of the architecture (fascicle length, muscle volume, physiological cross-sectional area, and anatomical cross-sectional area) of the lateral and medial gastrocnemius and the Achilles tendon moment arm was assessed using a technique that combined ultrasound imaging and motion analysis. Positive correlations were observed between the length ($r = 0.499$, $p = 0.049$) and size variables (muscle volume $r = 0.621$, $p = 0.010$; ACSA $r = 0.536$, $p = 0.032$) of the lateral gastrocnemius and the Achilles tendon moment arm, but correlations were only observed for size variables (muscle volume $r = 0.638$, $p = 0.008$; PCSA $r = 0.525$, $p = 0.037$; ACSA $r = 0.544$, $p = 0.029$), and not the length, of the medial gastrocnemius. These findings suggest lateral gastrocnemius adapts to moment arms to maintain force production throughout the range of motion across individuals, while the medial gastrocnemius does not and is thus better suited for static force generation.

## Introduction

Muscle strength, considered in terms of the moment produced by a muscle about a joint it spans, is attributable to both the force produced by the muscle and the muscle's leverage (*i.e.*, the moment arm of the tendon about the joint center or joint axis). Attention to how force and leverage not only combine but also interact with one another is critical to furthering

volume estimates, ultrasound videos on the Achilles tendon during calf raises for ATMA estimates, static ultrasound images of the LG and MG for muscle architecture estimates, MATLAB data files with summary data, and a README.txt file to orient the reader. These data are available in the Open Science Framework repository using the following citation with doi: Faux-Dugan L. Data Repository for "Correlations between Achilles Tendon Moment Arm and Plantarflexor Muscle Architecture Variables" [Internet]. OSF; 26 Jul 2024. doi:10.17605/OSF.IO/TH5GM.

**Funding:** The author(s) received no specific funding for this work.

**Competing interests:** The authors have declared that no competing interests exist.

knowledge of how human movement is either enabled or limited by muscle function. The ankle plantarflexor moment generated by the lateral and medial heads of gastrocnemius (LG and MG) and soleus, for example, has been studied with reference to factors presumed to influence the triceps surae forces, as well as in relation to the moment arm of the Achilles tendon (ATMA). These investigations often have been focused on associations between locomotor function and muscle force generating parameters [1, 2] or ATMA [3–5].

A muscle's moment arm about a joint directly determines the muscular moment generated about the joint for a given muscle force, but moment arm may also modulate the production of muscle force. Moment arm has a role in determining force because it is proportional to the muscle-tendon unit length change for a given joint rotation [6]. All else equal, muscle-tendon length changes will be less when the moment arm is smaller, and this reduction may lead to smaller changes in sarcomere lengths and slower sarcomere shortening velocities, each of which influence muscle force generating capacity through force-length and force-velocity effects [7, 8]. Nagano and Komura [9] used musculoskeletal computer simulations to demonstrate such interactions between ATMA and plantarflexor force during the generation of plantarflexor power and moment. They showed that shortening ATMA while maintaining plantarflexor optimal fiber length in a musculoskeletal model could reduce plantarflexor shortening velocity to such an extent that the loss of leverage was overcome by enhanced force production, resulting in greater joint moment and joint power during rapid plantarflexion movements. This surprising finding that muscular power and moment could be enhanced by reducing muscle moment arm depended on holding optimal fiber lengths constant; if the number of sarcomeres is reduced proportionally with moment arm, there would be no difference in fiber length changes with movement, and moment and power would be less due to the reduction in leverage.

The question of whether fiber (or fascicle) length is proportional to moment arm across individuals is important because a lack of proportionality would imply inter-individual differences in the capacity of specific muscles to generate force and, therefore, contribute to movement. For example, a muscle that has a smaller ratio of fiber length to moment arm should see a more rapid decrease in force during concentric contraction due to greater sarcomere shortening and more rapid sarcomere shortening velocity, making the muscle better suited to isometric force generation, if all else is equal [7, 8]. Conversely, a muscle that has a larger fiber length to moment arm ratio, whether that muscle is the corresponding muscle in a different individual or a different muscle in the same individual, is better suited to maintaining force during shortening [7, 8]. In this case, sarcomere shortening and shortening velocity are reduced due to the presence of more sarcomeres in series, less total shortening for a given change in joint angle, or a combination of both of these factors.

The limited evidence we have on how optimal muscle fiber length scales with muscle moment arm across individuals in humans suggests that these variables are not well correlated. Lieber et al. [10] found considerable variability across participants in intraoperative measurements of the rate of sarcomere shortening with joint angle in two human forearm muscles, a finding indicative of a lack of correlation between optimal fiber length and moment arm. Maganaris et al. [11], using fascicle length as a proxy for fiber length, failed to find significant correlations between triceps surae fiber lengths with ATMA, nor did they find significant correlations between fiber length and patellar tendon moment arm for two of the vasti muscles. Similarly, Infantolino et al. [12] found the ratio of optimal fascicle length to moment arm to vary widely across eight first dorsal interosseus specimens taken from cadavers. Son et al. [13] found no significant correlations between the optimal fascicle lengths of eight plantarflexors and the length of the calcaneus (a frequent proxy for ATMA) in cadaveric specimens. Lastly, Murray et al. [14] also found no correlation between optimal fascicle length and peak tendon

moment arm for most of the elbow muscles, also in cadavers. However, there was a strong correlation found for the brachioradialis and extensor carpi radialis longus, which are the two elbow muscles with the longest fibers.

Similarly, an association between muscle moment arm and muscle size would have implications for inter-individual variation in maximal joint moment and power. Maximal joint moment has been found to be strongly correlated with muscle size, whether size is characterized by muscle volume, physiological cross-sectional area (PCSA), or anatomical cross-sectional area (ACSA) [15, 16]. Muscle moment arm has also been found to correlate positively with maximal joint moment [17–21], but few studies have examined whether muscle size and moment arm are large in the same individuals, or whether such covariation of moment arm and muscle size is attributable to body size effects. Positive correlations have been found between the ACSA of triceps brachii with respect to their elbow extension moment arm that may occur due to differences in muscle size across individuals [22, 23]. Quadriceps volume has also been found to correlate with patellar tendon moment arm at the knee [20]. At the ankle, Baxter and Piazza [17] found a correlation between triceps surae muscle volume and ATMA (although the correlation did not reach levels of statistical significance, with $p = 0.054$), and Rasske et al. [5] noted greater ATMA at instants during the stance phase of walking when the plantarflexors were loaded, as compared to when the ankle was at the same angle but unloaded.

The purpose of the present study was to investigate further whether muscle length (fascicle length) and muscle size (muscle volume, PCSA, and ACSA) correlate with ATMA in untrained healthy young adults. We considered this relationship in two plantarflexor muscles, the LG and MG, which are known to differ in their architecture. LG tends to have a structure optimized for force production over a range of motion (with longer optimal fiber length and a smaller PCSA) and MG is better suited for isometric force production (with shorter optimal fiber length and greater PCSA) [8]. We hypothesized that, for both muscles, (1) fascicle length would correlate positively with ATMA, maintaining similar sarcomere operating ranges across individuals; and (2) muscle size variables (muscle volume, PCSA, and ACSA) would also correlate positively with ATMA because a larger muscle would tend to draw the tendon away from the ankle joint axis [3, 5, 22–25]. Recognizing body size as a potential confounding variable in the relationships between moment arm, fascicle length, and muscle size, we performed additional regression analyses that included body mass and height to determine whether moment arm independently explained variation in the dependent variables.

## Methodology

### Participants

Sixteen healthy young adult participants (9M/7F; stature 1.69 ± 0.09 m; body mass 70.30 ± 12.41 kg; age 27.8 ± 5.0 y) were recruited for this study. Individuals were excluded from the study if they were under 18 or above 45 years of age, had a significant history of lower limb musculoskeletal injuries or surgeries, neurological impairments, current ankle or foot pain, had a BMI greater than 30 kg m$^{-2}$, or competed in athletics at the college level or higher. All research study protocols were approved by the Institutional Review Board (STUDY00020090) at The Pennsylvania State University, and each participant provided informed consent prior to data collection.

### Anthropometric measures

Data collection was completed in a single testing session for each participant in which height and body mass were recorded, followed by measurement of shank and foot lengths with a tape

measure as the participant was standing in anatomical position. Shank length was the distance from the lateral tibial condyle to the lateral malleolus, and foot length was measured as the distance from the posterior aspect of the calcaneus to the anterior tip of the hallux.

## Ultrasound and motion system calibration and synchronization

Ultrasound imaging was used in conjunction with three-dimensional motion analysis to determine muscle volumes and ATMA using procedures described below. Prior to data collections, spatial calibration of the ultrasound probe was completed using the single-wall phantom procedure that has been previously described and validated by Prager et al. [26] and Stradwin software (v. 6.2, 64bit; Machine Intelligence Laboratory; Cambridge, UK). This probe calibration yielded homogeneous transformations between the ultrasound image coordinate system and a local probe coordinate system. This transformation was subsequently used to place the image in the global reference frame. Briefly, this procedure involved rigidly attaching cluster of four markers to the ultrasound probe. Using a pointer with markers attached, three virtual markers were then created that were fixed within the probe coordinate system and aligned with the probe imaging plane.

Ultrasound images were obtained using a 60 mm linear ultrasound probe (LV9.0/60/128Z-2) with a B-mode ultrasound system (30 Hz; LogicScan 128, Telemed, Lithuania; 64-bit Echo Wave II software, v. 4.1.0), synchronized with an 8-camera motion capture system (Eagle cameras; Cortex software v. 8.1.0.2017; Motion Analysis Corporation) recording marker coordinates at 100 Hz. Synchronization of motion data and ultrasound images was accomplished using a 5 V analog square wave output by the ultrasound beam former and sampled as an analog input to the data acquisition board of the motion system (National instruments PCIe-6259; Austin, TX). After data collection, marker trajectories were trimmed to be coincident with the ultrasound data, and the motion and ultrasound data were combined to create Stradwin input files, using a custom-written routine in MATLAB R2022b (Mathworks, Inc; Natick, MA), where a calibration procedure was carried out that minimized errors in phantom location in a least-squares sense.

## Muscle volume

The volumes of LG and MG were measured using B-mode ultrasonography images and the motion capture system. The participant lay prone on an examination table with their right foot extending past the end of the table and the dorsum of the foot aligned with end of table. The right leg was scanned for all participants, with the knee fully extended and the ankle in neutral position. A gelatin mold with a calf-shaped cavity was draped over the posterior aspect of the shank, covering the region beginning at the inferior margin of the tibial condyles to approximately two thirds of the way down the shank (Fig 1B). The participant was instructed to relax their muscles and remain as still as possible during the ultrasound scanning trials, and scanning trials were re-collected if participant movement was detected by the experimenter.

Two overlapping sets of approximately parallel, proximal-to-distal ultrasound image sweeps were collected for both the LG and MG, with ultrasound images of transverse cross-sections of each muscle being recorded at 30 Hz. Each sweep was completed in approximately 15–20 s. Most of the participants' muscles were imaged in two sweeps, but a third sweep was required for participants with larger muscles to ensure coverage of the entire muscle. Ultrasound gel was applied to the interfaces between the skin and gelatin mold and between the gelatin and the ultrasound probe. The entire image acquisition procedure was repeated five times for each muscle.

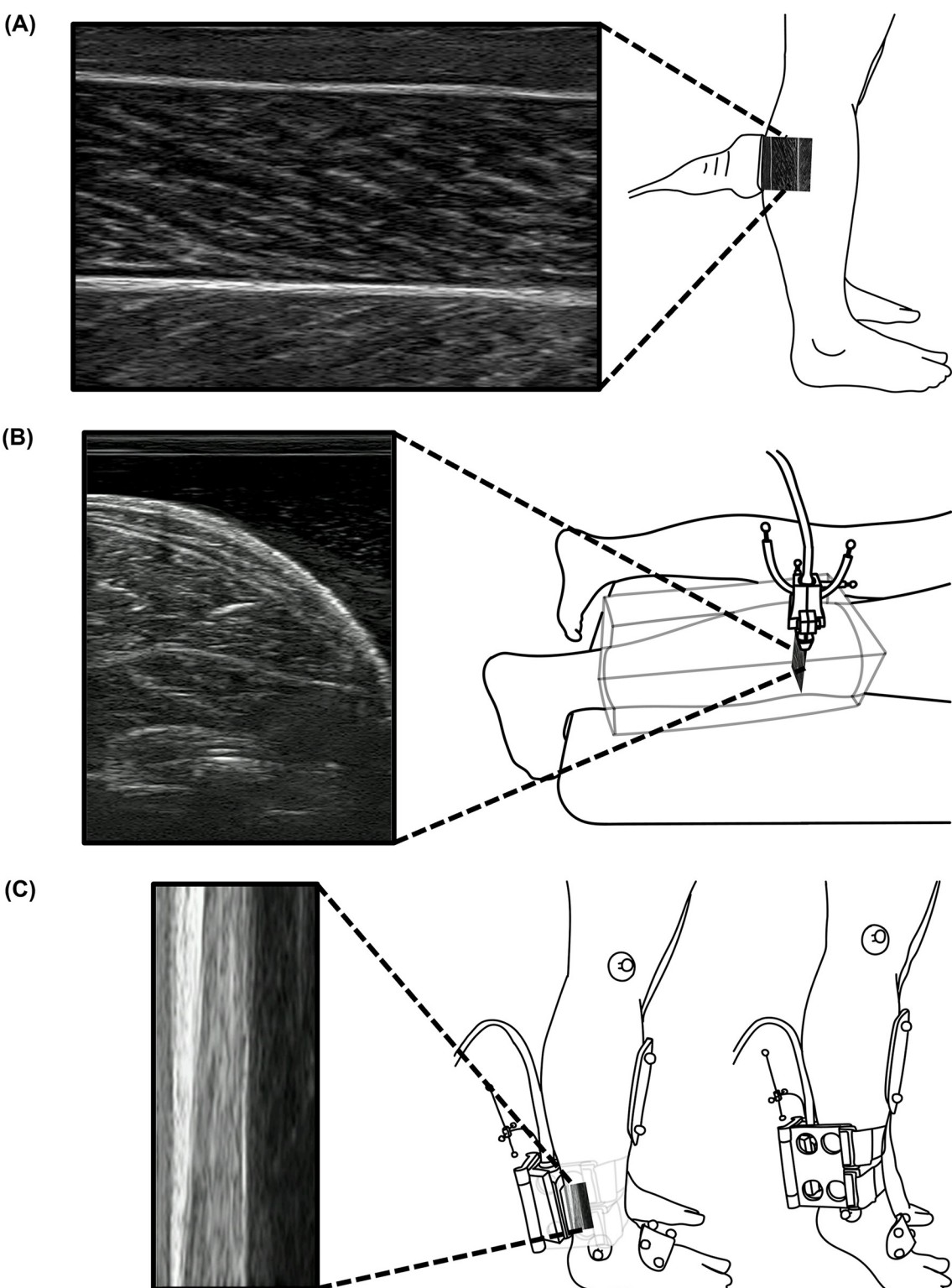

**Fig 1. Ultrasound image data collection sequence.** (A) Static quasi-sagittal plane ultrasound imaging of gastrocnemius. (B) Transverse ultrasound imaging of gastrocnemius for estimation of muscle volume. (C) Ultrasound imaging of the Achilles tendon along with tracking of the shank and foot for estimation of ATMA during toe-raise motions.

Marker trajectories and ultrasound images were then processed using a custom-written MATLAB script. Marker coordinates were filtered with a bidirectional, 4th order low-pass Butterworth filter with a 25 Hz cut-off frequency. Input files for Stradwin were created that contained the marker trajectories, local probe and ultrasound image coordinate systems (along with the homogenous transformation between these coordinate systems), ultrasound data specifications (*e.g.*, image resolution, video frame count, and image depth), and binary ultrasound video data. Linear interpolation of the 100 Hz motion capture data was used to align the motion data with the ultrasound image data that were captured at 30 Hz. Muscle boundaries were segmented to create contours (Fig 2A and 2C) separated by approximately 3 mm, at every fifth frame of the ultrasound images. All contours were manually determined by a single investigator (LFD). In Stradwin, maximal disc-guided shape-based interpolation was used to create three-dimensional models of the surfaces of both the LG and MG (Fig 2D). Muscle volumes were calculated in Stradwin from the contours using cubic planimetry integration algorithms described by Treece et al. [27]. Representative muscle volumes for each muscle were then computed by averaging three of the five model muscle volumes. The two trials most different from the median were excluded from this average, and models with muscle cross sections that were noticeably misaligned were not considered in the average calculation.

Separate tests of phantoms were completed to establish the reliability and accuracy of the volume estimates. Coefficients of variation for volume acquisition for phantoms of different sizes were all less than 0.5% and the average error was -1.9 ± 3.3%. Further details of these tests are found in S1 Appendix.

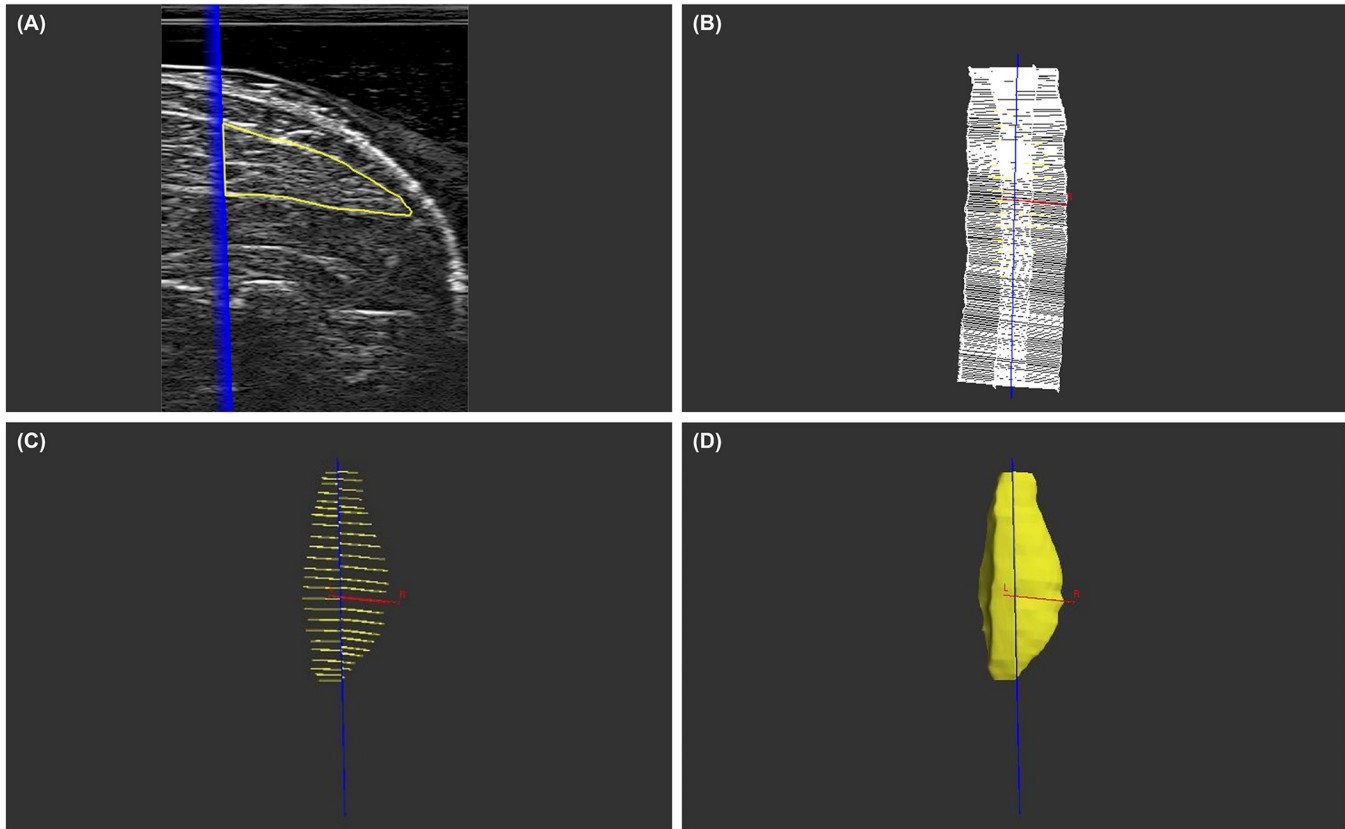

**Fig 2. Segmentation and volume acquisition.** (A) Contour created by segmenting LG boundaries. (B) Two overlapping, parallel stacks of planar ultrasound images that were taken along the length of the muscle. (C) Contours separated by approximately 3 mm along the length of the muscle. (D) Surface interpolated through the contours, from which muscle volume was computed.

## Muscle architecture parameters

Fascicle length and pennation angle were measured from quasi-sagittal plane static ultrasound images of the LG and MG. These images were collected as the participants stood in anatomical position with their knees and ankles in neutral position. Sets of five static ultrasound images were collected from the central region of the LG and MG (Fig 1A). The method described by Benard et al. [28] was followed to position the probe for these images: First, transverse images at the distal end of the muscle belly (approximately mid-shank) were made and the probe was adjusted until the deep aponeurosis was parallel to the bottom of the image. Then, the probe was rotated approximately 90˚ to align it with the mid-longitudinal axis and gradually moved proximally until the image was within the central region of the muscle (approximately two-thirds of the shank length up from the ankle), where the curvature of the fascicles and aponeuroses appeared to be minimized.

Each static ultrasound image of LG or MG was analyzed to calculate muscle architecture variables using a custom MATLAB script. The top and bottom extremes of each image were identified and the distance between these extremes in pixels was related to the scanning depth (in mm) to find a scaling factor (mm per pixel). Proximal and distal points were selected on the superficial and deep aponeuroses and on a representative fascicle (Fig 3). Muscle thickness was assessed proximally and distally as distance between the superficial and deep aponeurosis points along the image vertical direction [29, 30]. Muscle thickness, $t_m$, was then determined as the average of the proximal and distal thicknesses. Pennation angle, $\theta$, was calculated as the acute angle between the fascicle and the deep aponeurosis [29, 30]. Fascicle length, $L_f$, was calculated according to

$$L_f = \frac{t_m}{sin(\theta)} \tag{1}$$

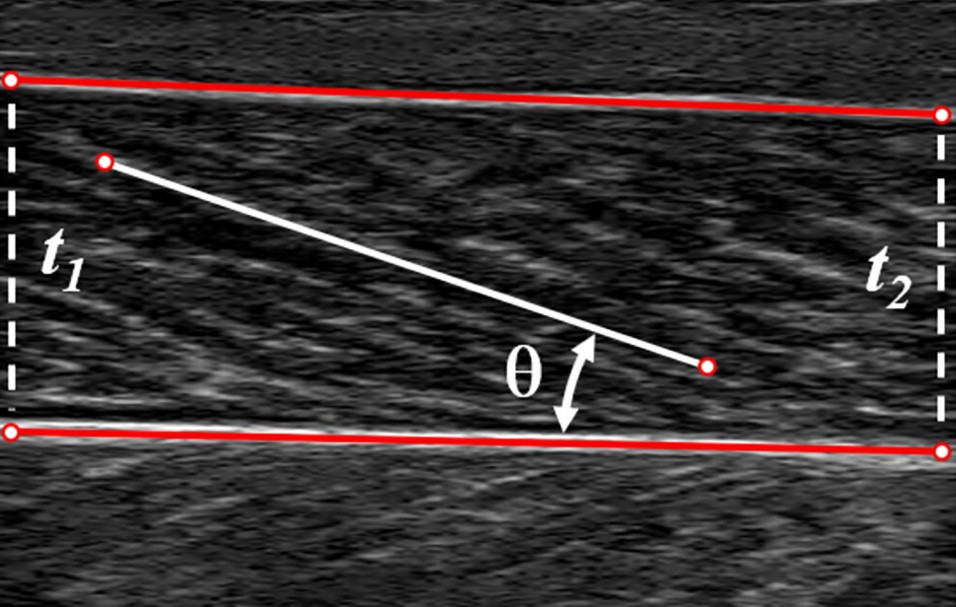

**Fig 3. Estimation of pennation angle and fascicle length from an ultrasound image.** A representative image of medial gastrocnemius is shown. Points were selected on the superficial and deep aponeuroses at the proximal and distal ends of the visible portion of the muscle and were used to estimate a proximal and distal thickness ($t_1$ and $t_2$). Muscle thickness, $t_m$, was the average of these two values. Pennation angle was the acute angle formed by the line segment representing the deep aponeurosis and a line segment formed by two points selected on a muscle fascicle. Fascicle length, $L_f$, was computed from thickness and pennation angle according to Eq 1.

Fascicle lengths and pennation angles were estimated in this manner five times for each image and an average fascicle length was taken.

Distances in pixels were then scaled to determine fascicle lengths in millimeters or centimeters. ACSA was estimated by calculating the maximum cross-sectional area from the cross-section contours of the muscle of interest. *PCSA* was calculated as the quotient of muscle volume, $V_m$, and average fascicle length, $L_f$, multiplied by the cosine of the pennation angle, $\theta$ [31]:

$$PCSA = \frac{V_m}{L_f} * cos(\theta) \qquad (2)$$

## Achilles tendon moment arm

Following the procedures previously described by Wade et al. [32], ATMA was measured by combining ultrasound imaging with three-dimensional motion analysis (using the same motion analysis system described above), but with some differences. A low-profile ultrasound probe (LV7.5/60/128Z-2) and custom designed 3D-printed plastic probe holder were used that more effectively held the scanning system in place over the tendon, in a standardized location approximately 2 to 3 cm above the malleoli. Following imaging of the Achilles tendon and functional determination of the ankle axis from the relative motion of the foot and shank during weight-bearing calf raises [32], two maximally separated proximal and distal points along the tendon line of action were found in every ultrasound image frame corresponding to four angles of interest: 0° (neutral position) and 5°, 10°, and 15° of plantarflexion. This was done by locating sagittal plane midpoints between the superficial and deep boundaries of the Achilles tendon in a custom-written MATLAB routine. A tendon midline was created connecting the tendon boundary midpoints, and this was taken to represent the tendon line of action. The coordinates of the tendon boundary midpoints in the image coordinate system were scaled from pixels to mm and transformed from the image coordinate system into the global frame. Finally, the ATMA about the ankle axis was calculated as described by Wade et al. [32]: First the moment of the unit force vector along the tendon about a point on the ankle axis was found, and then the length of the projection of that moment vector onto the ankle axis was computed.

## Statistical analysis

Simple linear regressions were performed to find correlations between ATMA (assessed at neutral ankle position) and the following muscle architecture parameters for each of LG and MG: fascicle length, muscle volume, PCSA, and ACSA. Statistical significance was defined using a threshold of $\alpha = 0.05$.

The influence of body-size variables was investigated using a forward-backward stepwise linear regression procedure to determine a multiple regression model for each LG and MG architecture response variable. The starting model contained the predictor variables ATMA along with body height, body mass, and the interactions between ATMA and height and mass. A predictor was added to the model if the p-value of the F-statistic was less than 0.05 and removed if the p-value was greater than 0.1. When ATMA was present among the predictor variables in the final model, this was taken to indicate that variation in ATMA contributes to the response variable in a manner that extends beyond body size. This analysis was carried out using the MATLAB function *stepwiselm()*.

## Results

Mean ATMA in neutral ankle position was 47.5 ± 9.3 mm and was found to increase with plantarflexion (Table 1). Mean fascicle length was substantially greater for LG, and pennation angle, PCSA, and ACSA were greater for MG (Table 2).

**Table 1. Means and standard deviations (n = 16) of Achilles tendon moment arm at 0˚, 5˚ plantarflexion (PF), 10˚ plantarflexion, and 15˚ plantarflexion of the lateral gastrocnemius and medial gastrocnemius.**

| ankle angle (degrees) | Achilles tendon moment arm (mm) |
|---|---|
| **0˚** | 47.5 ± 9.3 |
| **5˚ PF** | 49.1 ± 9.4 |
| **10˚ PF** | 50.2 ± 9.7 |
| **15˚ PF** | 51.7 ± 9.5 |

A significant moderate correlation was found between fascicle length and ATMA measured in neutral ankle position for LG (r = 0.499; p = 0.049), but the correlation between MG fascicle length and ATMA was not significant (r = 0.229; p = 0.393) (Fig 4). Similar significant (or nearly significant) and non-significant correlations were found between LG and MG fascicle length and ATMA assessed at the other ankle positions considered: 5˚ PF, 10˚ PF, and 15˚ PF. The same is true for all the correlations reported below, so we present the correlations involving ATMA measured in neutral position here and the remaining correlation results are presented in S2 Appendix.

Significant moderate correlations were also found between all MG size variables (volume, PCSA, and ACSA) and ATMA, and between two of the three LG size variables and ATMA. MG volume was correlated with ATMA (r = 0.638; p = 0.008), as were MG PCSA (r = 0.525; p = 0.037) and MG ACSA (r = 0.544; p = 0.029) (Figs 5–7). For LG, similar moderate correlations were found between LG volume and ATMA (r = 0.621; p = 0.010) and between LG ACSA and ATMA (r = 0.536; p = 0.032), but significant correlation was not found between LG PCSA and ATMA (r = 0.330; p = 0.212) (Figs 5–7).

ATMA was found to correlate moderately with both height (r = 0.574; p = 0.020) and body mass (r = 0.633; p = 0.009) in our sample. Stepwise regressions that included body-size variables produced models that in many cases, however, retained ATMA as a significant predictor of LG and MG architecture variables (Table 3). ATMA was the only remaining significant predictor of LG fascicle length and MG muscle volume, and it was retained along with body mass in the models for LG muscle volume and LG PCSA. For both LG muscle volume and LG PCSA, the term representing the interaction between body mass and ATMA was retained in the model, although the resulting model for LG PCSA only trended toward significance (p = 0.080). Interaction plots for these two models showed that the association between each outcome variable and ATMA depended on body mass in very different ways (Fig 8). When body mass was held fixed at a value at the upper end of the range of our sample, there was a positive dependence of each variable on ATMA, but for a small body mass, each varied

**Table 2. Means and standard deviations for fascicle length, pennation angle, muscle volume, physiological cross-sectional area (PCSA), and anatomical cross-sectional area (ACSA) of the lateral gastrocnemius and medial gastrocnemius.**

| | lateral gastrocnemius | | | medial gastrocnemius | | |
|---|---|---|---|---|---|---|
| outcome variable | all (n = 16) | M (n = 9) | F (n = 7) | all (n = 16) | M (n = 9) | F (n = 7) |
| **fascicle length (mm)** | 79.0 ± 14.9 | 82.0 ± 15.0 | 75.2 ± 15.0 | 60.3 ± 9.0 | 58.9 ± 7.7 | 62.3 ± 10.8 |
| **pennation angle (˚)** | 10.9 ± 2.7 | 11.5 ± 2.9 | 10.1 ± 2.6 | 18.9 ± 3.4 | 20.2 ± 3.2 | 17.3 ± 3.1 |
| **muscle volume (cm³)** | 128.0 ± 41.0 | 146.5 ± 35.0 | 104.3 ± 37.5 | 203.2 ± 56.8 | 234.4 ± 44.9 | 163.0 ± 45.2 |
| **PCSA (cm²)** | 16.8 ± 6.4 | 18.9 ± 7.2 | 14.1 ± 4.1 | 34.3 ± 9.8 | 40.2 ± 7.1 | 26.6 ± 7.1 |
| **ACSA (cm²)** | 10.6 ± 2.7 | 11.9 ± 2.6 | 8.9 ± 2.1 | 14.4 ± 3.0 | 16.1 ± 2.3 | 12.2 ± 2.4 |

Results are presented for all participants, and for the male and female subgroups.

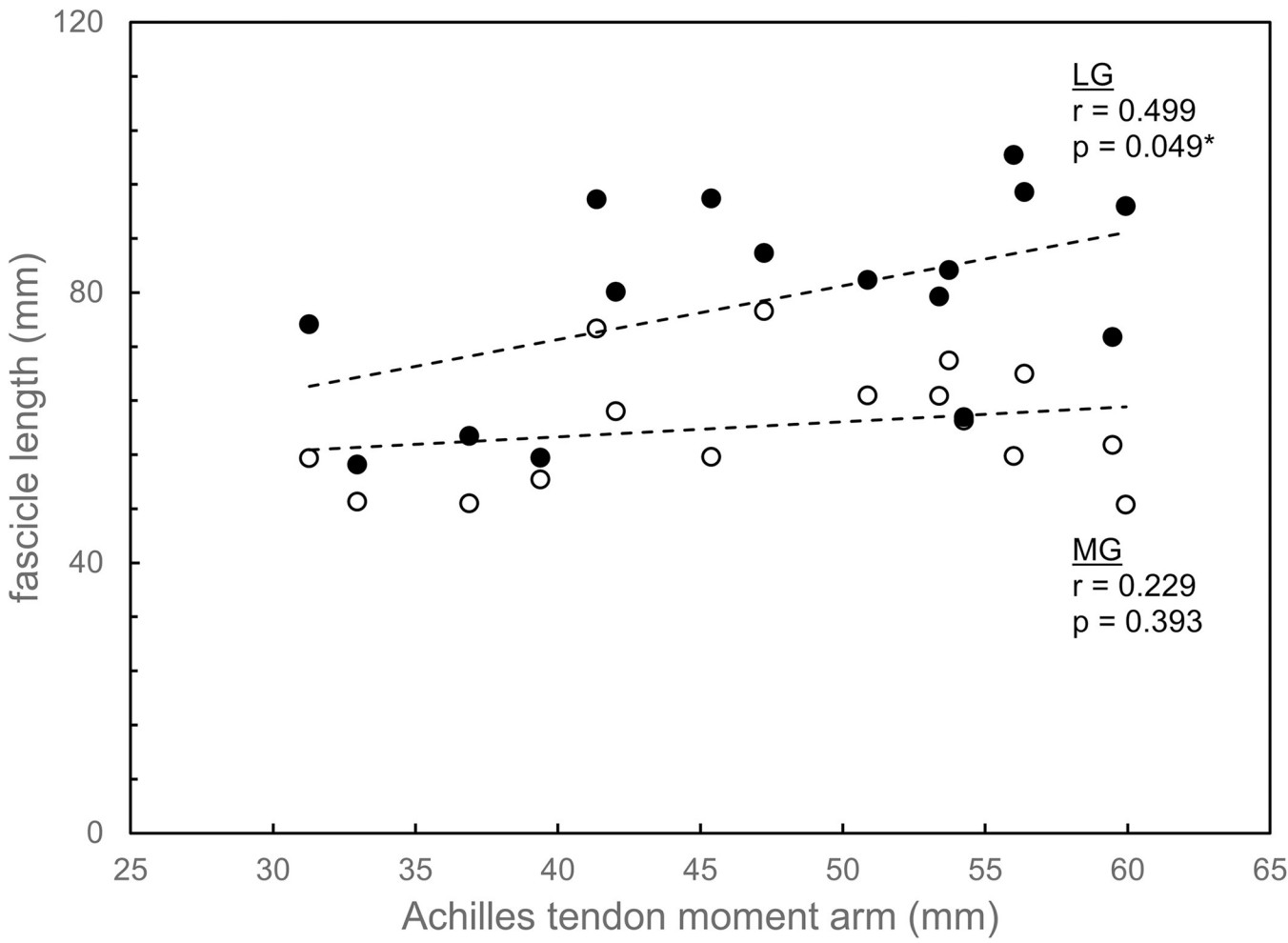

**Fig 4. Fascicle lengths of the lateral gastrocnemius (LG; filled circles) and medial gastrocnemius (MG; open circles) plotted versus Achilles tendon moment arm as assessed in neutral ankle position.** Linear regression lines indicate correlations between each muscle's fascicle length and moment arm. An asterisk (*) denotes a significant correlation at the α = 0.05 level.

negatively with ATMA. Only body mass and/or body height were retained in the models for MG PCSA and ACSA for both muscles.

## Discussion

Among the key findings in this study is that significant positive correlations were found between fascicle length and ATMA for LG but not for MG (Fig 4). All muscle size variables (muscle volume, PCSA, and ACSA) were correlated with ATMA, with the exception of PCSA for LG (Figs 5–7). These correlations could not be entirely attributed to the influence of body size, as ATMA was retained in several of the stepwise regression models (Table 3), suggesting that some of the variation in LG fascicle length, muscle volume, and PCSA, as well as MG PCSA were explained by ATMA.

The findings of this study partially supported the hypotheses. The hypothesis that there would be a positive correlation between fascicle length and ATMA was supported for LG but not for MG (Fig 4). Muscle size variables were generally positively correlated with ATMA for both muscles, supporting the second hypothesis, but some of these correlations seemed to be due to body-size effects (Figs 5–7; Table 3).

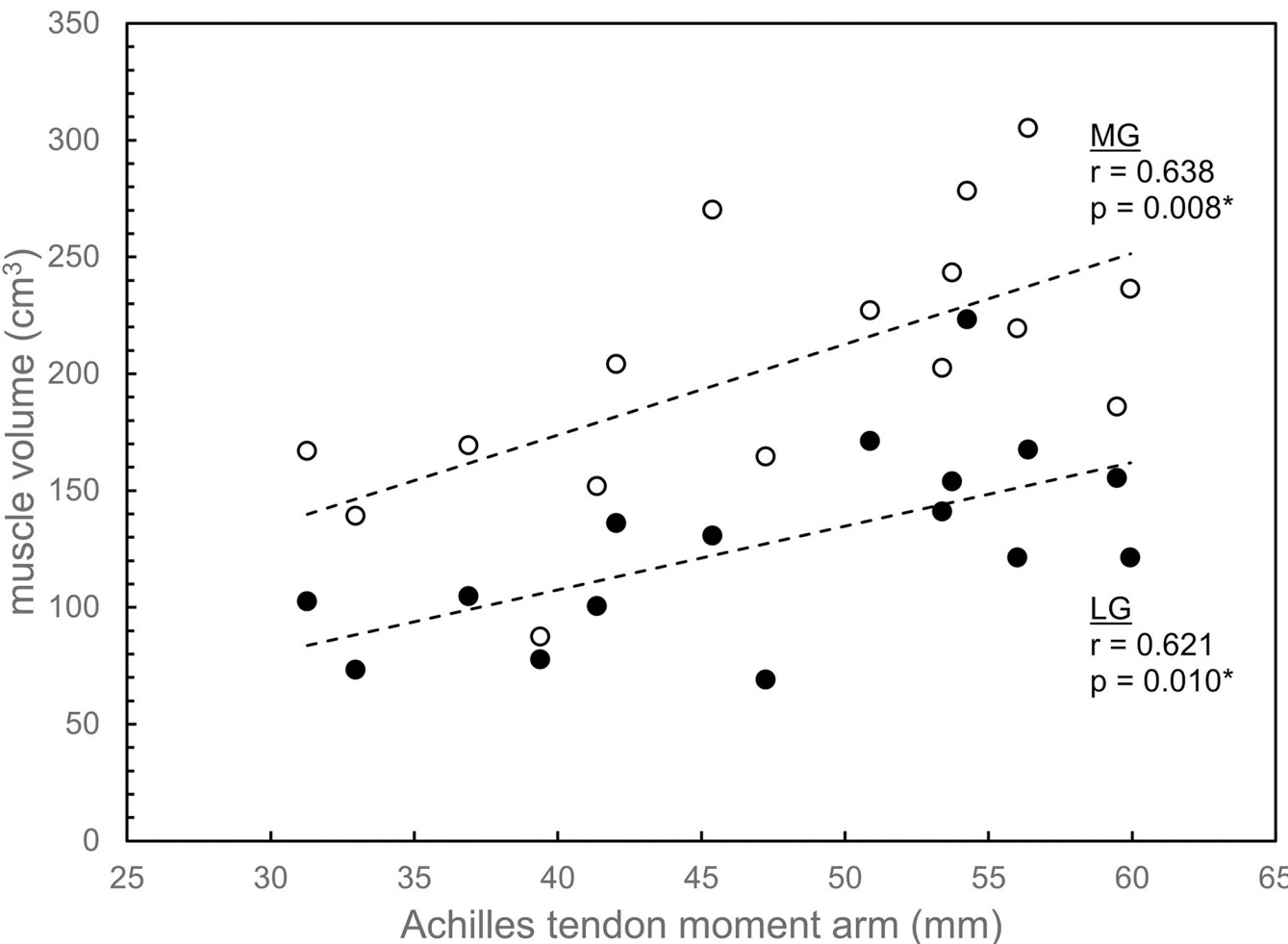

**Fig 5. Muscle volumes for lateral gastrocnemius (LG; filled circles) and medial gastrocnemius (MG; open circles) plotted versus Achilles tendon moment arm.** Linear regression lines indicate correlations between each muscle's volume and moment arm. An asterisk (*) denotes a significant correlation at the α = 0.05 level.

Fascicle lengths of the LG (79.0 ± 14.9 mm) and MG (60.3 ± 9.0 mm) were comparable to values previously reported by Fouré et al. [33] (LG, 91.0 ± 31.0 mm; MG, 58.0 ± 11.0 mm), Maganaris et al. [21] (LG, 74.0 ± 3.4 mm; MG, 45.0 ± 2.3 mm), and Crouzier et al. [34] (LG, 67.0 ± 8.0 mm; MG, 59.0 ± 7.0 mm). Additionally, the average relative differences between the muscle volumes (LG and MG) of the present study and other studies were ~19% and ~14%, respectively [34–38]. The differences observed between the muscle volumes of the present study and previous studies might be explained by factors such as the predominance of male subjects in previous studies (often >80%), unlike the present study (9 of 16 male), and differences in methods for muscle volume estimation.

Our results also generally agree with those from previous investigations of the relationship between tendon moment arm and muscle geometry across joint systems. Similar to the present study, Baxter and Piazza [17] reported a positive correlation between triceps surae muscle volume and ATMA, although this relationship only approached significance (p = 0.054). In contrast to the results of the present study, Maganaris et al. [11] observed no correlation either between plantarflexor fascicle length and Achilles tendon moment arm, or between quadriceps fascicle lengths and patellar tendon moment arm measured *in vivo*. Sugisaki et al. [22] and

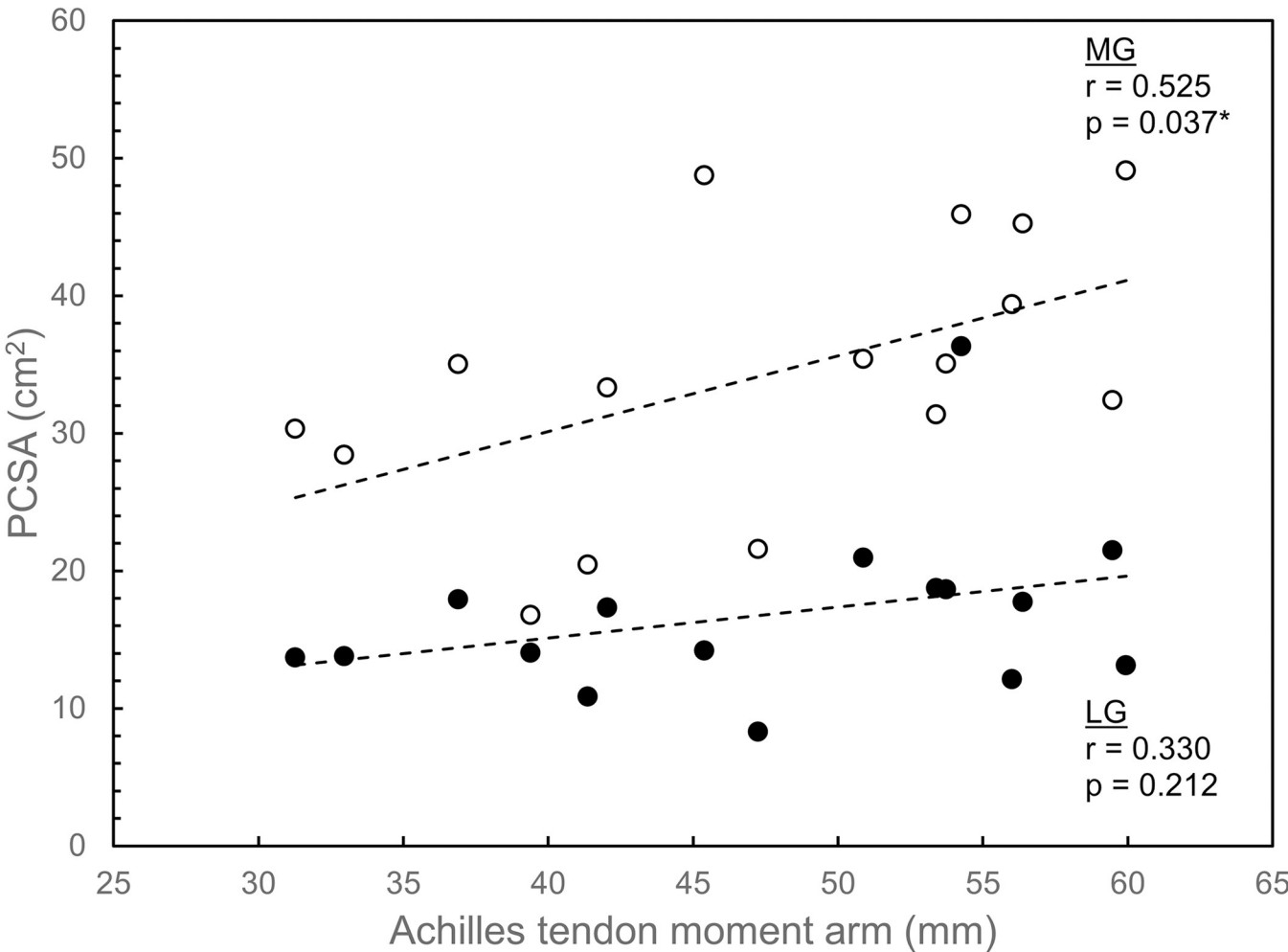

**Fig 6. Physiological cross-sectional area (PSCA) for lateral gastrocnemius (LG; filled circles) and medial gastrocnemius (MG; open circles) plotted versus Achilles tendon moment arm.** Linear regression lines indicate correlations between each muscle's PSCA and moment arm. An asterisk (*) denotes a significant correlation at the $\alpha = 0.05$ level.

Sugisaki et al. [23] reported positive correlations between triceps brachii anatomical cross-sectional area (ACSA) and tendon moment arm, findings that correspond to the correlations between ACSA and ATMA for both LG and MG found in the present study. Additionally, using a cadaver model, Murray et al. [14] reported strong, significant positive correlations between optimal fascicle lengths and peak tendon moment arms across several elbow muscles. Across specimens for the same muscle, however, these correlations were observed only for the brachioradialis and extensor carpi radialis longus. Interestingly, these were the two muscles with the smallest cross-sectional area and longest fibers, which is also characteristic of LG (for which a correlation was found between fascicle length and moment arm in the present study) relative to MG (for which no such correlation was found). These findings suggest that fascicle length may scale more closely with moment arm in muscles that have long fibers and small size.

The correlations across individuals between fascicle length and moment arm observed in this study for LG and by Murray et al. [14] for longer-fibered elbow muscles suggest an architectural arrangement by which reductions in muscle force generation are moderated when joint excursions are large. When individuals with longer ATMA for have correspondingly

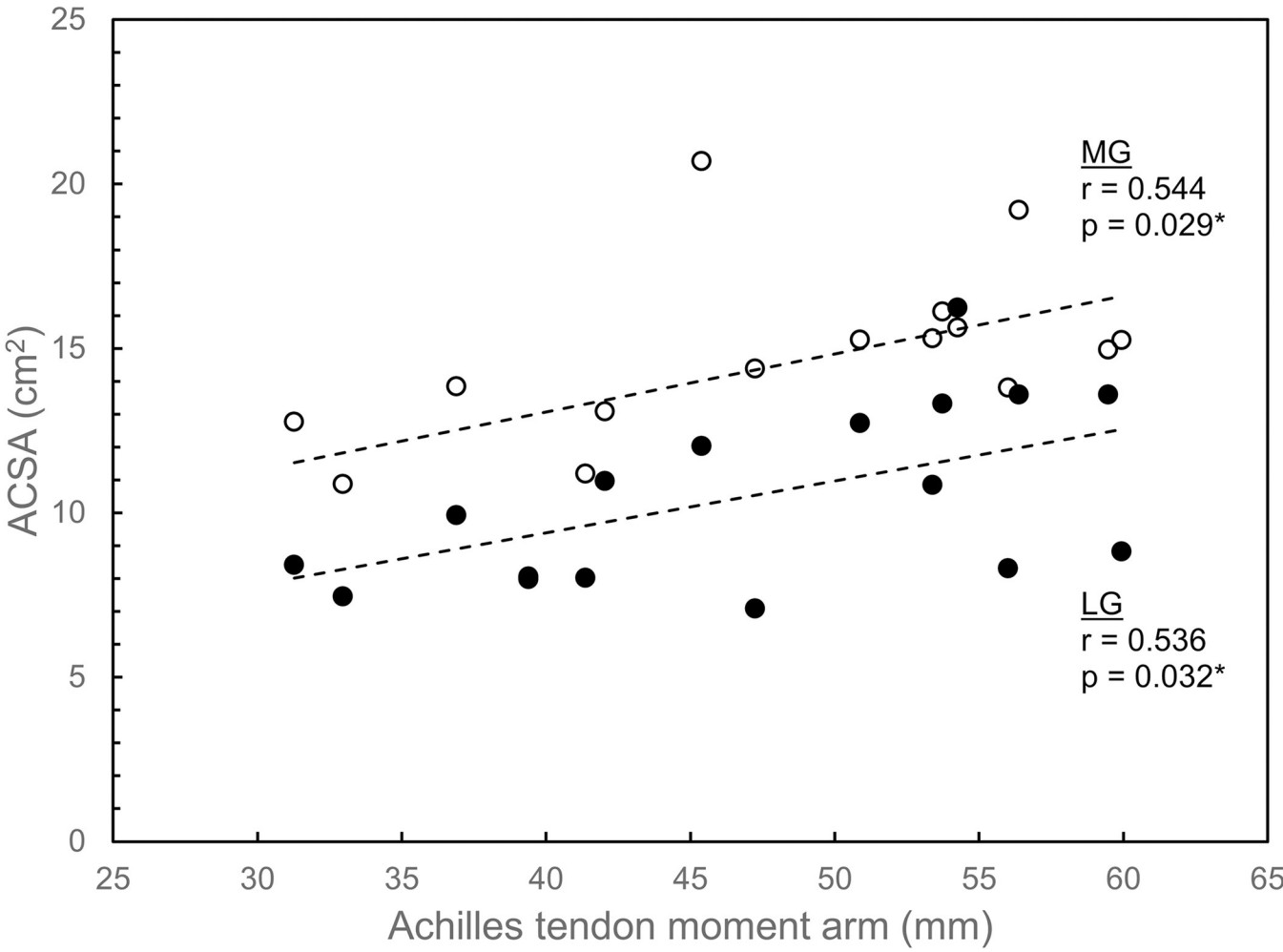

**Fig 7. Anatomical cross-sectional area (ASCA) for lateral gastrocnemius (LG; filled circles) and medial gastrocnemius (MG; open circles) plotted versus Achilles tendon moment arm.** Linear regression lines indicate correlations between each muscle's ASCA and moment arm. An asterisk (*) denotes a significant correlation at the $\alpha = 0.05$ level.

**Table 3. Regression models predicting lateral gastrocnemius and medial gastrocnemius architecture variables that were produced by a stepwise regression procedure.**

| outcome variable | lateral gastrocnemius | | | medial gastrocnemius | | |
|---|---|---|---|---|---|---|
| | model | $r^2$ | p | model | $r^2$ | p |
| fascicle length (mm) | 41.2 + 0.80 ATMA | 0.249 | 0.049* | N/A | N/A | N/A |
| muscle volume (cm$^3$) | 439.2–5.47 M – 11.1 ATMA + 0.177 M*ATMA | 0.629 | 0.006* | 18.2 + 3.89 ATMA | 0.408 | 0.008* |
| PCSA (cm$^2$) | 93.5–1.12 M – 2.42 ATMA + 0.034 M*ATMA | 0.419 | 0.080 | -1.99 + 0.459 M | 0.340 | 0.018* |
| ACSA (cm$^2$) | 0.82 + 0.139 M | 0.393 | 0.009* | -16.9 + 18.5 H | 0.319 | 0.023* |

This procedure included body-size variables body mass (M) and body height (H) in addition to Achilles tendon moment arm (ATMA). For the medial gastrocnemius fascicle length, no predictors except the constant term remained in the model following the stepwise procedure. An asterisk

(*) denotes a significant difference between the resulting model and the constant model at the $\alpha = 0.05$ level.

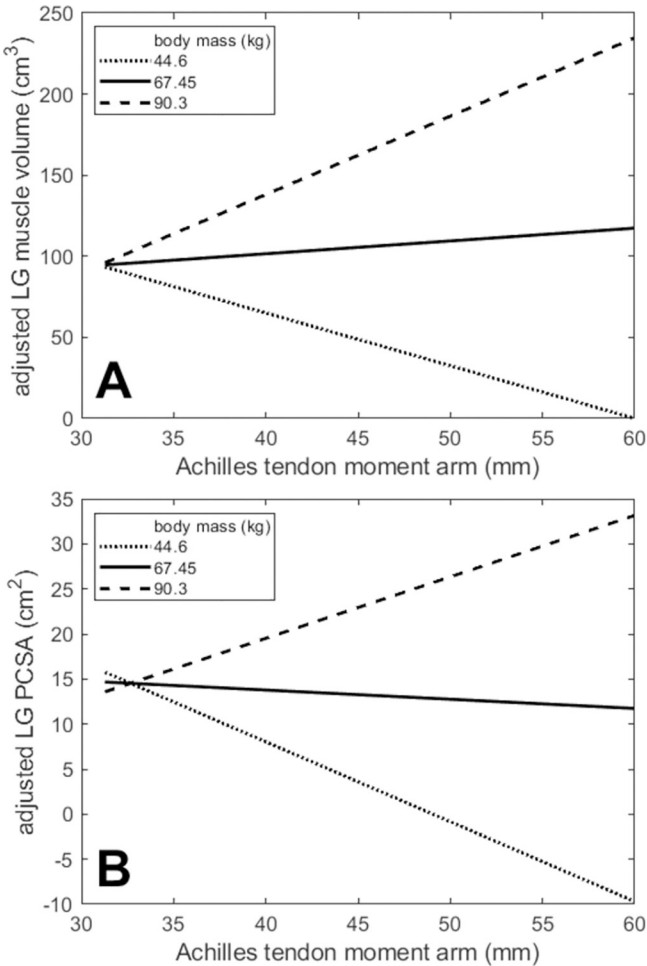

**Fig 8. Interaction plots for the two models determined using stepwise regression that retained the term representing the interaction between body mass and Achilles tendon moment arm.** Lines show the model-predicted dependence of (A) lateral gastrocnemius muscle volume and (B) lateral gastrocnemius physiological cross-sectional area (PCSA) on Achilles tendon moment arm for three fixed values of body mass.

longer LG fascicles, sarcomere operating ranges will be similar across individuals and thus the loss of force generating capacity (due to more rapid sarcomere shortening velocities) that might occur with shorter fascicles and longer moment arms is mitigated. It is notable that LG fascicle length was best predicted by ATMA alone, even when body size is taken into account (Table 3). Further study is needed to understand how LG contributions to plantarflexor moment depend on body size; the interactions we found suggest that individuals with higher body masses rely on greater moment arms and LG muscle size in a way that smaller individuals do not (Fig 8).

While we did find correlations between muscle size and ATMA for MG, it appears that these relationships may be attributable to body size. No correlations were found between MG fascicle length and ATMA, nor between MG fascicle length and height or body mass. Muscles like MG, with a relatively large PCSA and short fibers, may be better suited for contributing to postural stabilization [7, 8] rather than for dynamic force production during shortening over large length changes, compared to muscles like LG that have longer fibers. It is important to note, however, that muscle contributions to function during movement are determined by many factors beyond the architecture parameters considered here, including muscle

composition, motor unit recruitment, and differences in how movements are performed. An accurate assessment of how muscle architecture contributes to function would need to take such factors into account.

The correlations we found between ACSA (for both LG and MG) and ATMA could be seen as evidence of larger plantarflexor muscles having longer ATMAs due to inter-individual differences in muscle size previously noted for both the triceps surae [5] and for the elbow extensors [22]. We did not find the ACSA of either muscle to be correlated with ATMA independent of body size, however. This finding suggests that larger individuals may have larger muscles that tend to draw the tendon path away from the joint. Maganaris et al. noted a similar effect when they described increases in ATMA that accompanied triceps surae contraction [24].

There are certain limitations to this study. Though the present protocol for ultrasound of muscle architecture is based on the protocols used in prior studies [28, 30, 39, 40], errors may arise when using this method. Processing images when fascicles extend beyond the ultrasound field of view, or when fascicular and aponeurotic curvature is excessive, can lead to inaccurate estimations of fascicle lengths [41]. Additionally, potential errors in fascicle length estimations can come from tissue deformation due to applying excessive probe surface pressure to superficial tissues [38] and misaligning the ultrasound probe with the fascicular plane during imaging [28]. Muscle architecture was also measured only in the central region to estimate average fascicle length for a muscle, but muscle geometry varies by region of measurement. However, the central region is often used to capture muscle architecture since fascicle lengths tend to be more homogenous in this region [40], and the fascicle lengths of the present study were comparable to those published previously (see above). Furthermore, fascicle length was used as a proxy for fiber length, and therefore sarcomere lengths, but sarcomere lengths tend to vary by fibers across muscle region and joint angle [42–45]. However, in the present study, fascicle lengths were estimated, across participants, within approximately the same joint angle (e.g., neutral ankle angle) and region of the muscles (e.g., central). Recent evidence suggests an association between fascicle length and sarcomere lengths by region of the muscle and joint angle [42], and serial sarcomere numbers tend to adapt in response to typical mechanical loading patterns to optimize force at routinely expressed joint angles [46, 47]. Lastly, we acknowledge that some of our measurements of muscle structure were made with the muscles at rest (muscle volumes and ACSA) while other were made with the muscles under load (fascicle length and pennation angle during standing; ATMA during weight-bearing calf raises). Values measured for ATMA have been shown to depend on the contractile state of the triceps surae [5, 24], and it is possible that this mismatch could have affected our results. This concern is mitigated somewhat because skeletal muscle volume changes between rest and contraction have been demonstrated to be negligible [48] and because all other measures except ACSA are made under some form of load.

The findings of the present study suggest avenues for further study on how plantarflexor muscle architecture adapts to ATMA and on the determinants of muscular plantarflexor moments. Our finding that fascicle length is positively correlated with ATMA for LG suggest a preservation of muscle length and velocity operating ranges across individuals that should be investigated further using techniques that permit assessment of sarcomere lengths. We did not find an association between fascicle length and ATMA for MG, and future work should consider whether this architecture, seemingly well suited to static stabilization, prevents MG from contributing positive work during dynamic shortening tasks in some individuals. The positive correlations found between size variables (for both LG and MG) and ATMA are similar to recent evidence that differences in muscle size is associated with variation in ATMA [5, 22, 23]. The present results identifying relationships between architecture variables cannot say

anything directly about how these associations come about, whether it is through training history, genetic determinants, or other factors. Further study is needed using approaches that permit control of such factors, such as animal models.

## Supporting information

**S1 Appendix. Accuracy and reliability of three-dimensional ultrasound volume estimations.**
(DOCX)

**S2 Appendix. Summary of the results of regression models not presented in the main part of the paper.**
(DOCX)

## Acknowledgments

The authors wish to thank Rebecca Gonzalez, BS, for her assistance during the data collections.

## Author Contributions

**Conceptualization:** Logan Faux-Dugan, Stephen J. Piazza.

**Data curation:** Logan Faux-Dugan.

**Formal analysis:** Logan Faux-Dugan.

**Investigation:** Logan Faux-Dugan, Stephen J. Piazza.

**Methodology:** Logan Faux-Dugan, Stephen J. Piazza.

**Project administration:** Stephen J. Piazza.

**Software:** Logan Faux-Dugan.

**Supervision:** Stephen J. Piazza.

**Writing – original draft:** Logan Faux-Dugan.

**Writing – review & editing:** Stephen J. Piazza.

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
