## [Decision Letter · Decision Letter 0]

21 Jun 2024

PONE-D-24-15521Correlations between Achilles Tendon Moment Arm and Plantarflexor Muscle Architecture VariablesPLOS ONE

Dear Dr. Piazza,

Thank you for submitting your manuscript to PLOS ONE. After careful consideration, we feel that it has merit but does not fully meet PLOS ONE’s publication criteria as it currently stands. Therefore, we invite you to submit a revised version of the manuscript that addresses the points raised during the review process.

**I believe that with some thoughtful editing, taking the suggestions of the reviewers into account, this manuscript will be acceptable for publication. **

We look forward to receiving your revised manuscript.

Kind regards,

Charlie M. Waugh

Academic Editor

PLOS ONE

Journal Requirements:

Reviewers' comments:

Reviewer's Responses to Questions

**Comments to the Author**

1. Is the manuscript technically sound, and do the data support the conclusions?

Reviewer #1: Yes

Reviewer #2: Yes

2. Has the statistical analysis been performed appropriately and rigorously? 

Reviewer #1: Yes

Reviewer #2: Yes

3. Have the authors made all data underlying the findings in their manuscript fully available?

Reviewer #1: No

Reviewer #2: Yes

4. Is the manuscript presented in an intelligible fashion and written in standard English?

Reviewer #1: Yes

Reviewer #2: Yes

5. Review Comments to the Author

**Reviewer #1: **This manuscript sets out to quantify the presence of correlations between the Achilles tendon moment arm and length and size variables of the medial and lateral head of the gastrocnemius, while also investigating the modifying influence of body anthropometrics. The experimental methods are rigorous, the results are clearly described, and the manuscript is generally well written. Weaknesses in the current version of the manuscript include a lack of a well-defined gap in the literature, a description of the need for these data, confusion surrounding the notion of muscle bulging that is not resolved until late in the discussion, and an incomplete study purpose to include the role of anthropometrics. I appreciated the opportunity to review.

Major Concerns

1. The most important consideration with this manuscript is that the gap and therefore need for the resultant data are never clearly defined. Rather, a lengthy summary of highly related outcomes are introduced prior to stating the study purpose. I encourage a more thoughtful description of the scientific and or translational gap in our understanding left by these past references and the ways in which this particular study is designed to address them.

2. The purpose is clearly stated but I believe incomplete. Brief mention of covariation being explained by body size effects is included in the introduction. These factors them become quite significant in the results and discussion section. However, these analyses and their intended goal are absent from the study purpose and hypotheses. I recommend addressing more explicitly up front.

3. There is major confusion over the use of the term bulging which caused this reviewer to believe there were significant methodological problems with the stated hypotheses, analyses, and interpretations. These are ultimately resolved by the end of the discussion, however, I recommend avoiding the term entirely. Rather, my understanding is that the authors simply mean inter-individual differences in muscle size. Recommend going with something closer to that throughout.

4. I tend to believe the results of the paper are overinterpreted and some of the conclusions too strong. Specifically, implying that the gastrocnemius is better suited for static force generation and thereby questioning whether it can perform positive mechanical work during functional tasks. I believe the available evidence in the literature confirms that the gastrocnemius does preform positive mechanical work, for example during walking. So – more nuance is needed to avoid misrepresenting the intended conclusions. In addition, are there alternative interpretations that could be

Minor Concerns

Methods

L 219. Check for grammatical mistake here.

L265. Please provide a definition for the ankle axis.

L267. Were these ATMA values estimated at rest? Please disclose and discuss any relevant implications.

Discussion

L402-406. The authors provide an “interesting” observation but provide no discussion about why this is interesting and what it means. Recommend addressing. In addition, the end of this paragraph ends with restating prior associations that were outlined in the introduction. Similar to other comments, there is an enormous need here to contextualize the current findings in the gap left by this prior work.

L420-425. This section is pivotal to the authors conclusions but is incredibly brief and underdeveloped. I would encourage much more thought be put into elaborating on these findings and their implications to ensure the conclusions are sound.

Data availability. I was unable to locate the underlying data in the Penn State Data Commons. More direct link with description in the manuscript document would be helpful.

**Reviewer #2: **Generally, the manuscript is excellent and I thoroughly enjoyed reading it. I only have a very few minor comments and questions.

Line 123 - There is an error in your stature units – this should be m not cm according to your values.

Volume measures – out of interest, why did you take volume measures lying down at rest, but all other measures when standing? Would this have affected your correlations at all?

ATMA - At what point/location of the Achilles tendon were measurements obtained? Over the free tendon? Was/how was this standardised between participants?

Figures – I assume this is the online image compressor, but figure quality appears poor, so may be worth checking. E.g., in figure 1 a and b, it is difficult to see the contract of US images.

6. PLOS authors have the option to publish the peer review history of their article (what does this mean?). If published, this will include your full peer review and any attached files.

Reviewer #1: **Yes: **Jason R. Franz

Reviewer #2: No

---

## [Author Response · Author response to Decision Letter 0]

26 Jul 2024

Note to Both Reviewers

We thank both reviewers and the editor for their consideration of the initial submission. Careful revisions have been made to the manuscript to address the concerns of both reviewers. The Introduction has been revised to better highlight the gap in our current understanding that is addressed by our work. Efforts were also made to avoid overstating our conclusions in the Discussion. Lastly, several methodological limitations were defined and addressed.

Please note that in preparing our revision, we discovered a minor problem with the description of how muscle volumes were computed by averaging across multiple trials. This has been corrected in the revised manuscript (L201 – L204).

Below are the responses to each reviewer, with the reviewers’ comments followed by our response. Where we cite line numbers for changes to the manuscript, they refer to the “clean” copy of the revised manuscript.

Response to Reviewer #1 

This manuscript sets out to quantify the presence of correlations between the Achilles tendon moment arm and length and size variables of the medial and lateral head of the gastrocnemius, while also investigating the modifying influence of body anthropometrics. The experimental methods are rigorous, the results are clearly described, and the manuscript is generally well written. Weaknesses in the current version of the manuscript include a lack of a well-defined gap in the literature, a description of the need for these data, confusion surrounding the notion of muscle bulging that is not resolved until late in the discussion, and an incomplete study purpose to include the role of anthropometrics. I appreciated the opportunity to review.

We thank Reviewer #1 for the careful consideration of our work, for the positive assessment of our manuscript, and for the suggestions for improvement. We believe that the changes we made in response to these comments will make the paper a much better addition to the literature.

Major Concerns

1. The most important consideration with this manuscript is that the gap and therefore need for the resultant data are never clearly defined. Rather, a lengthy summary of highly related outcomes are introduced prior to stating the study purpose. I encourage a more thoughtful description of the scientific and or translational gap in our understanding left by these past references and the ways in which this particular study is designed to address them.

We agree with Reviewer #1 that the rationale for the study was not presented clearly enough. As we see it, the gap in our current knowledge is that we do not know whether basic muscle architecture variables such as fiber length and muscle size scale with muscle moment arm across individuals. Many authors (ourselves included) have noted that moment arms have the potential to affect muscle contributions to movement not only by determining leverage but also by setting muscle operating points on F-L and F-V curves. If fiber lengths scale with moment arm, however, these operating points will be constant across individuals. Similarly, if muscle size (as a proxy for force-generating capacity), is negatively correlated with moment arm, then moment-generating capacity is not different across individuals. We regretfully note that none of this came across very well in our manuscript; we briefly identified the question in the third paragraph of the Introduction and then reviewed several studies before expanding on the importance of the question in the fourth paragraph. We now see that it would be much better to clearly frame both the question and its importance before reviewing the relevant studies.

In response to the Reviewer’s insightful comment, we have done a major reorganization of the Introduction along these lines in the revised manuscript, with respect to both fiber-length and muscle size scaling. We now clarify the gap and the need before the literature review, which will help the reader to understand better why we have done the study. Please note that paragraphs 3-5 of the Introduction have been rearranged and revised in the revision, with the former fourth paragraph explaining importance now moved up to become the third paragraph (L67 – L107).

2. The purpose is clearly stated but I believe incomplete. Brief mention of covariation being explained by body size effects is included in the introduction. These factors them become quite significant in the results and discussion section. However, these analyses and their intended goal are absent from the study purpose and hypotheses. I recommend addressing more explicitly up front.

We thank the Reviewer for bringing this to our attention, and agree that the analyses incorporating body size should be included when the statement of our study’s purpose is made. We have now included these analyses, along with the reason for their inclusion, in the final paragraph of the Introduction (L118 – L121).

3. There is major confusion over the use of the term bulging which caused this reviewer to believe there were significant methodological problems with the stated hypotheses, analyses, and interpretations. These are ultimately resolved by the end of the discussion, however, I recommend avoiding the term entirely. Rather, my understanding is that the authors simply mean inter-individual differences in muscle size. Recommend going with something closer to that throughout.

We agree with the Reviewer’s observation regarding “bulging” and have followed the Reviewer’s recommendation that we simply refer to muscle size instead. We have made this change everywhere “bulging” appeared in the manuscript (L23; L102; L117; L435-438; L475).

4. I tend to believe the results of the paper are overinterpreted and some of the conclusions too strong. Specifically, implying that the gastrocnemius is better suited for static force generation and thereby questioning whether it can perform positive mechanical work during functional tasks. I believe the available evidence in the literature confirms that the gastrocnemius does preform positive mechanical work, for example during walking. So – more nuance is needed to avoid misrepresenting the intended conclusions. In addition, are there alternative interpretations that could be

We agree with the Reviewer that some of our conclusions were stated too strongly. With respect to the specific example mentioned above, we did not mean to suggest that medial gastrocnemius is not suited for positive mechanical work, although that is essentially what we wrote. The revised sentence now represents what we originally intended to convey, which was that our results may indicate a more limited capacity for dynamic force generation in MG relative to LG (L425-428).

The Reviewer’s last comment appears to be cut off, but we believe we understand the Reviewer’s point that alternative explanations should be considered. In response we have acknowledged that factors aside from the architecture parameters considered here could be at play in determining muscle function during movement (L428-432).

Minor Concerns

Methods

L 219. Check for grammatical mistake here.

In response to the Reviewer’s comment, we have changed “participant” to “participants”. We hope that this has addressed the problem (L168).

L265. Please provide a definition for the ankle axis.

We thank the Reviewer for the opportunity to clarify this point. In our determination of moment arm, we used a “functional” ankle axis determined from finite helical axes that are computed from the relative motion of the foot and shank during the heel raise motions. These calculations and procedures are described in detail by Wade et al. [1]. In the revised manuscript, we have briefly stated how the ankle axis was located and provided a reference to this paper (L265 – L266).

L267. Were these ATMA values estimated at rest? Please disclose and discuss any relevant implications.

In response to the Reviewer’s question, we have stated that moment arms were calculated from data collected during “weight-bearing calf raises”, and referenced Wade et al. (2019) (L266). As suggested by the Reviewer, and in response to a question from Reviewer #2, we mention that this methodological choice potentially creates a mismatch in our data collection (L458), with variables being assessed under different conditions. We do not know enough about the implications of this mismatch to offer anything more than speculation to the reader in terms of how we expect our results might differ. In the limitation section of the Discussion of the revised manuscript, we have nevertheless acknowledged that the measures were made under different conditions and that it may have affected our results, along with describing mitigating factors that lessen this concern (L463 – L465). Specifically, we noted that skeletal muscle volume changes between rest and contraction are vey small, and that almost all the other measures (with the exception of ACSA) are made under some form of load.

Discussion

L402-406. The authors provide an “interesting” observation but provide no discussion about why this is interesting and what it means. Recommend addressing. In addition, the end of this paragraph ends with restating prior associations that were outlined in the introduction. Similar to other comments, there is an enormous need here to contextualize the current findings in the gap left by this prior work.

In response to the Reviewer’s comment, we have streamlined and clarified this section of the Discussion by making several changes. First, we have moved the mention of the Maganaris et al. [2] study, whose results contrast with our own, closer to the beginning of the paragraph (L394 – L411). A sentence has been added to the end of the same paragraph to clarify what we found interesting about the correspondence to the previous results of Murray et al. [3], and this sentence also serves as a transition to the following paragraph, in which the implications are explored in more detail.

L420-425. This section is pivotal to the authors conclusions but is incredibly brief and underdeveloped. I would encourage much more thought be put into elaborating on these findings and their implications to ensure the conclusions are sound.

We agree with the Reviewer that this short paragraph on findings related to MG is important to our conclusions, and we see it as fitting within a larger section of the Discussion that covers the implications of our finding related to scaling with reference to the architecture of both LG and MG. To address the Reviewer’s concern, this paragraph has been expanded upon in the revised manuscript (L423 – L432), and other structural changes have been made to clarify this part of the Discussion (please see our response to the previous comment).

Data availability. I was unable to locate the underlying data in the Penn State Data Commons. More direct link with description in the manuscript document would be helpful.

We apologize for the confusion on this point. At the time of the original submission of this paper, the data were being prepared for being placed online. The data are now online at the Penn State Data Commons, accessible via the following URL:

https://doi.org/10.26208/HDQA-DB53

In the decision letter, however, we learned that the Penn State Data Commons “does not qualify as an acceptable data repository according to PLOS's standards.” We were advised to upload a minimal necessary dataset to one of PLOS’s recommended repositories, and we have done this also. The data may also be found in the Open Science Framework repository using the following citation with doi:

Faux-Dugan L. Data Repository for “Correlations between Achilles Tendon Moment Arm and Plantarflexor Muscle Architecture Variables” [Internet]. OSF; 26 Jul 2024. doi:10.17605/OSF.IO/TH5GM

Response to Reviewer #2

Generally, the manuscript is excellent and I thoroughly enjoyed reading it. I only have a very few minor comments and questions.

We would like to thank the Reviewer #2 for their kind words and positive feedback on our work.

Line 123 - There is an error in your stature units – this should be m not cm according to your values.

Thank you – the correction has been made (L125).

Volume measures – out of interest, why did you take volume measures lying down at rest, but all other measures when standing? Would this have affected your correlations at all? 

This concern is an important one that was also raised by Reviewer #1. Please see our full response above, but, briefly, in the revised manuscript, we acknowledge that the different measures were made under different conditions and that there was the potential for this mismatch to have affected our correlations, although we cannot say precisely how. We also point out that volumes are likely to be very similar between rest and contraction conditions, and that almost all our measures were made with the muscle under some form of loading (L458 – L465).

ATMA - At what point/location of the Achilles tendon were measurements obtained? Over the free tendon? Was/how was this standardised between participants?

For each subject, the ultrasound probe was affixed at a location approximately 2-3 cm above the malleoli. We thank the Reviewer for bringing this point of clarification to our attention, and in the revised manuscript, we make the location of the measurements of the Achilles tendon clear (L264).

Figures – I assume this is the online image compressor, but figure quality appears poor, so may be worth checking. E.g., in figure 1 a and b, it is difficult to see the contract of US images.

We agree with the Reviewer that the image quality in the PDF is poor. As the Reviewer suggests, this appears to be due to compression of the images that was done in creating the PDF. The quality of the images we uploaded is much better, and it appears that those original images are accessible by clicking on the link above the figure in the PDF that reads “Click here to access/download;Figure;Fig1.tif”. We hope that this resolves the Reviewer’s concern.

References Cited

1. Wade FE, Lewis GS, Piazza SJ. Estimates of Achilles tendon moment arm differ when axis of ankle rotation is derived from ankle motion. J Biomech. 2019;90:71-7.

2. Maganaris CN, Baltzopoulos V, Tsaopoulos D. Muscle fibre length-to-moment arm ratios in the human lower limb determined in vivo. J Biomech. 2006;39(9):1663-8.

3. Murray WM, Buchanan TS, Delp SL. The isometric functional capacity of muscles that cross the elbow. J Biomech. 2000;33(8):943-52.

---

## [Decision Letter · Decision Letter 1]

13 Aug 2024

Correlations between Achilles tendon moment arm and plantarflexor muscle architecture variables

PONE-D-24-15521R1

Dear Dr. Piazza,

We’re pleased to inform you that your manuscript has been judged scientifically suitable for publication and will be formally accepted for publication once it meets all outstanding technical requirements.

Kind regards,

Charlie M. Waugh

Academic Editor

PLOS ONE

Additional Editor Comments (optional):

Reviewers' comments:

Reviewer's Responses to Questions

**Comments to the Author**

1. If the authors have adequately addressed your comments raised in a previous round of review and you feel that this manuscript is now acceptable for publication, you may indicate that here to bypass the “Comments to the Author” section, enter your conflict of interest statement in the “Confidential to Editor” section, and submit your "Accept" recommendation.

Reviewer #1: All comments have been addressed

Reviewer #2: All comments have been addressed

2. Is the manuscript technically sound, and do the data support the conclusions?

Reviewer #1: Yes

Reviewer #2: (No Response)

3. Has the statistical analysis been performed appropriately and rigorously? 

Reviewer #1: Yes

Reviewer #2: (No Response)

4. Have the authors made all data underlying the findings in their manuscript fully available?

Reviewer #1: Yes

Reviewer #2: (No Response)

5. Is the manuscript presented in an intelligible fashion and written in standard English?

Reviewer #1: Yes

Reviewer #2: (No Response)

6. Review Comments to the Author

Reviewer #1: The authors have addressed all of my comments and the manuscript is very much improved. I appreciate the author's effort and congratulate them on a nice paper.

Reviewer #2: Thank you to the authors for their detailed responses to the review. I am happy that they have addressed my concerns/questions, and I have no further comments.

7. PLOS authors have the option to publish the peer review history of their article (what does this mean?). If published, this will include your full peer review and any attached files.

Reviewer #1: **Yes: **Jason R. Franz

Reviewer #2: No

---

## [Editor Report · Acceptance letter]

19 Aug 2024

PONE-D-24-15521R1 

PLOS ONE

Dear Dr. Piazza, 

I'm pleased to inform you that your manuscript has been deemed suitable for publication in PLOS ONE. Congratulations! Your manuscript is now being handed over to our production team.

Kind regards, 

on behalf of

Dr. Charlie M. Waugh 

Academic Editor

PLOS ONE